# Dementia with Lewy Bodies: Genomics, Transcriptomics, and Its Future with Data Science

**DOI:** 10.3390/cells13030223

**Published:** 2024-01-25

**Authors:** Thomas R. Goddard, Keeley J. Brookes, Riddhi Sharma, Armaghan Moemeni, Anto P. Rajkumar

**Affiliations:** 1Mental Health and Clinical Neurosciences Academic Unit, Institute of Mental Health, School of Medicine, University of Nottingham, Nottingham NG7 2TU, UK; 2Department of Biosciences, School of Science & Technology, Nottingham Trent University, Nottingham NG11 8NS, UK; 3Biodiscovery Institute, School of Medicine, University of Nottingham, Nottingham NG7 2RD, UK; 4UK Health Security Agency, Radiation Effects Department, Radiation Protection Science Division, Harwell Science Campus, Didcot, Oxfordshire OX11 0RQ, UK; 5School of Computer Science, University of Nottingham, Nottingham NG8 1BB, UK

**Keywords:** dementia, Lewy bodies, genomics, transcriptomics, data science, machine learning

## Abstract

Dementia with Lewy bodies (DLB) is a significant public health issue. It is the second most common neurodegenerative dementia and presents with severe neuropsychiatric symptoms. Genomic and transcriptomic analyses have provided some insight into disease pathology. Variants within *SNCA*, *GBA*, *APOE*, *SNCB*, and *MAPT* have been shown to be associated with DLB in repeated genomic studies. Transcriptomic analysis, conducted predominantly on candidate genes, has identified signatures of synuclein aggregation, protein degradation, amyloid deposition, neuroinflammation, mitochondrial dysfunction, and the upregulation of heat-shock proteins in DLB. Yet, the understanding of DLB molecular pathology is incomplete. This precipitates the current clinical position whereby there are no available disease-modifying treatments or blood-based diagnostic biomarkers. Data science methods have the potential to improve disease understanding, optimising therapeutic intervention and drug development, to reduce disease burden. Genomic prediction will facilitate the early identification of cases and the timely application of future disease-modifying treatments. Transcript-level analyses across the entire transcriptome and machine learning analysis of multi-omic data will uncover novel signatures that may provide clues to DLB pathology and improve drug development. This review will discuss the current genomic and transcriptomic understanding of DLB, highlight gaps in the literature, and describe data science methods that may advance the field.

## 1. Introduction

Dementia is a clinical syndrome that encompasses several neurodegenerative disorders [1]. It is a progressive cognitive and functional decline beyond that which is expected in typical ageing [1]. Dementia is characterised by the 11th revision of the International Classification of Diseases as a marked decline in two or more cognitive domains relative to an individual’s previous cognitive functioning and age [2]. Global estimates suggest that at least 50 million individuals are currently diagnosed with dementia, and this syndrome was the seventh leading cause of global mortality in 2019 [3,4]. Dementia is also the leading global cause of disability and dependency among older individuals [1]. The prevalence of dementia doubles every 5 years after the age of 65 years [5], and it is projected that 152 million individuals will be living with dementia by 2050 [3]. The economic burden will also magnify, and it has been suggested that the global cost of dementia has surpassed USD 1 trillion and is likely to reach USD 2 trillion by 2030 [6,7]. The increasing prevalence, coupled with the significant socio-economic impacts, demonstrates that dementia is a prominent health concern that requires immediate attention.

Dementia with Lewy bodies (DLB) is a common type of neurodegenerative dementia [8], second in prevalence only to Alzheimer’s disease (AD) [9]. Meta-analyses show that DLB accounts for 4.2% of dementia cases diagnosed in community settings and 7.5% within secondary care [9]. This is likely a significant underestimate, and neuropathological examination studies indicate that DLB may account for up to 20% of dementia cases [10]. People with DLB experience a faster rate of cognitive decline, a shorter life expectancy, a greater cost of care, and a greater prevalence of neuropsychiatric symptoms [11]. DLB has four core clinical features (fluctuating cognition, recurrent visual hallucinations, rapid eye movement sleep behaviour disorder (RBD), and spontaneous parkinsonism), of which at least two are required to provide clinical diagnosis [12].

DLB is a primary synucleinopathy, as α-synuclein aggregation is the key initial step in the formation of Lewy bodies and Lewy neurites, the pathological hallmarks of DLB [13,14]. In a pathogenic state, *α*-synuclein has been shown to aggregate and combine with at least 90 distinct molecules to form Lewy bodies and Lewy neurites [13,14], which spread throughout the brain in a prion-like manner [15]. DLB can be separated into three subtypes depending on the distribution of Lewy pathology within the brain [15]. Brainstem-predominant DLB refers to Lewy pathology primarily located within the substantia nigra and the locus coeruleus, and it presents with a greater prevalence of RBD [16]. Neocortical (diffuse) DLB refers to Lewy pathology within the cerebral cortex, with or without the presence of Lewy bodies in the brainstem, and it is more closely associated with cognitive decline [16,17]. Limbic (transitional) DLB refers to pathology within the anterior cingulate and transentorhinal cortices, and this typically has a longer disease duration [16,18].

However, the process of Lewy body and Lewy neurite formation and its relationship to disease pathology remains unclear. As such, specific disease-modifying treatments for DLB are not available, and the typical survival time from diagnosis is approximately 4 years [19,20]. Moreover, the poor molecular understanding of DLB causes inaccurate diagnosis. Existing diagnosis relies largely upon clinical observations, as there are no reliable biological fluid-based biomarkers available for DLB. The sensitivity of clinical diagnoses is estimated to be as low as 25% [21], and recent studies have suggested that the diagnostic rate can vary two-fold between clinicians [12,21,22]. Current indicative biomarkers that rely on imaging methods or polysomnography, such as the detection of reduced dopamine transporter activity in the corpus striatum [23], are promising yet often not feasible in most mental health settings in the UK [24]. As such, nearly 50% of people with DLB may remain misdiagnosed as AD or another dementia [24]. There is a clear need for research that advances the molecular understanding of DLB so that therapeutic targets and diagnostic biomarkers may be identified to improve clinical diagnosis and management of DLB. 

Genomic and transcriptomic analysis has provided insight into the molecular pathology of DLB, and previous reviews have summarised these findings [25,26,27]. This review will provide an update on recent analyses within the field, identifying research gaps that remain, and reveal how more sophisticated data science methodologies may be used to fill these gaps.

## 2. The Genetics of DLB

Most genomic association studies within DLB have focussed on candidate genes that have been implicated within other dementias or synucleinopathies [26]. The first genome-wide association study (GWAS) within the field, which was conducted in 2017, precipitated the widespread identification of genetic associations and has facilitated significant advancements in the field [28]. Subsequent GWASs and other genetic investigations have identified and validated variants that may be implicated in DLB [26,29]. A list of replicated genetic associations is presented in Table 1.

### 2.1. SNCA

Synucleinopathies, including DLB, are associated with the aggregation of *α*-synuclein. The gene for *α*-synuclein, *SNCA* (chr4q22), has therefore been researched extensively. Candidate-gene studies have identified significant associations within the locus, specifically common intronic variants rs974711 and rs1348224, and the risk of DLB [30,31]. Subsequent GWASs support this association and discovered additional significant variants, including two upstream variants in the gene, rs7681440 and rs2301135 [28,29]. Although there are variants within *SNCA* that do not associate with DLB [39], the connection between *SNCA* and DLB is well established [27,31,39,40,41]. 

However, the role of the *SNCA* variants within DLB pathology is unclear. It has been hypothesised that these variants increase the propensity of *α*-synuclein to aggregate and inhibit membrane binding activity [27]. Further investigation is needed to understand the functional outcomes of these variants.

### 2.2. GBA

Variants within the glucosylceramidase beta gene (*GBA*; chr1q22) have also been consistently associated with DLB [32,42]. The *GBA* gene encodes the lysosomal glucosylceramidase enzyme, which is responsible for the degradation of *α*-synuclein [26,27]. Candidate-gene studies have identified multiple variants, including rare missense variants rs2230288, rs76763715, and rs368060, that occur in less than 1% of all individuals, which increase the risk of DLB between two- and ten-fold [32,42,43]. Subsequent GWASs validated these findings and detected additional variants within *GBA* that significantly associated with DLB incidence [28,29]. 

Whilst the link between *GBA* variants and DLB has been established, the pathogenesis of these variants is poorly understood [27]. Individuals with *GBA* variants have been associated with an earlier onset of DLB and a shorter life expectancy [32]. It has been hypothesised that *GBA* variants impede the production of glucosylceramidase, reducing the degradation of *α*-synuclein within lysosomes [27]. Without sufficient degradation, *α*-synuclein accumulates and aggregates, precipitating DLB pathology [27]. Interestingly, a recent investigation identified that loss-of-function mutations in *GBA* were associated with reduced levels of *α*-synuclein in cerebrospinal fluid (CSF) [44]. Reduction in CSF *α*-synuclein levels may be attributable to an accumulation of *α*-synuclein within the brain due to reduced clearance from the brain parenchyma [44].

### 2.3. APOE

The apolipoprotein E (*APOE*) gene, located on chromosome 19q13, is the most replicated genetic association within dementia research and within DLB [26]. The *APOE* gene is associated with cholesterol transportation in the brain, as its product combines with lipids to form lipoproteins [45]. There are three major isoforms of APOE, which are determined based upon the genotypes of coding single nucleotide polymorphisms (SNPs) rs429358 and rs7412 [45,46]. A recent meta-analysis discovered that 21 out of the 25 studies on APOE-ε4 and DLB displayed statistically significant associations [26]. The aggregated risk for DLB in individuals with the APOE-ε4 alleles was nearly three-fold (*p* < 0.001) [26]. The meta-analysis primarily included candidate-gene studies, but one GWAS, which discovered and replicated statistical association, was also included within the meta-analysis [26,28]. Since publication of the meta-analysis, APOE-ε4 has been significantly associated with DLB in another GWAS and one more candidate-gene study [29,47]. 

The APOE-ε4 isoform is associated with a greater risk of dementia, and it is considered to promote the aggregation of amyloid-*β* (A*β*) [45,48]. A*β* aggregation is a common pathological feature of several dementias, and it is also detected sporadically within DLB brains [49,50]. The APOE-ε4 isoform may be associated with DLB through nonamyloidogenic mechanisms [51]. Fragments of APOE-ε4 are neurotoxic and may promote neurodegeneration through disruption to the cytoskeleton and impairment of mitochondrial function [51]. Research has also identified that APOE-ε4 promotes synucleinopathies independent of A*β*, possibly through alterations in lipid metabolism and synaptic function [52,53]. Given that synucleinopathies are a predominant pathological feature of DLB, this finding may partly explain the association between APOE-ε4 and DLB.

### 2.4. SNCB and SNCG

Two paralogs of *SNCA* with conserved N-terminal domains, *SNCB* and *SNCG*, have been associated with DLB [34]. The *SNCB* and *SNCG* genes translate to *β*-synuclein and *γ*-synuclein, respectively, distinct forms of synuclein that were previously thought to not be associated with DLB [34]. Rare missense variants within *SNCB* (chr5q35) at codons 70 (V70M) and 123 (P123H) have been detected in unrelated subjects [35]. Authors of the study did not detect these substitutions in any of the control samples and suggested that the variants may predispose individuals to DLB [35]. Three intronic *SNCB* variants also showed statistical association with DLB when compared to pathologically confirmed controls [34]. Five mutations within *SNCG* (chr10q23) have also displayed significant association with DLB [34]. Two of the *SNCG* variants were intronic, two were upstream mutations, and one, rs760113, was a missense mutation whereby the alternative allele was protective against DLB [34]. However, associations within *SNCB* and *SNCG* have not been replicated in two separate GWASs [28,54] and require further investigation.

It has been suggested that *β*-synuclein is an anti-aggregation agent antagonistic towards *α*-synuclein, and mutations within *SNCB* lead to a loss of function [55]. The role of *SNCG* is comparatively unclear. *γ*-synuclein regulates cytoskeletal remodelling and may influence DLB through this pathway [55], although membrane binding of both *β*-synuclein and *γ*-synuclein has been shown to form inclusion and induce toxicity [56]. Additional research is required to understand the role that these paralogs of *SNCA* play within DLB pathology. 

### 2.5. MAPT

The *MAPT* gene (chr17q21) encodes the microtubule-associated protein tau and has also been the focus of numerous candidate-gene studies [57,58,59]. The H1 haplotype, one of the two most common haplotypes within *MAPT*, has been associated with DLB [58,59]. Rare missense variants upstream of the repeat region within *MAPT*, A152T and G86S, have also been associated with DLB [37,59]. However, a recent GWAS did not detect a significant association between the *MAPT* locus and DLB [28], and a study that included pathologically diagnosed cases did not detect a significant association between the H1 haplotype and DLB [36].

The *MAPT* variants likely contribute to the hyperphosphorylation and aggregation of tau into neurofibrillary tangles. These aggregates are known to precipitate Lewy body formation and DLB pathology [60,61]. The H1 haplotype refers to the direct orientation of *MAPT*, which increases the expression of transcripts with four repeats (4R) [62]. 4R *MAPT* transcripts are associated with elevated hyperphosphorylation and aggregation of tau [63]. The A152T variant creates a phosphorylation site that contributes to the hyperphosphorylation of tau [36,59], whilst the functional consequence of G86S is unknown [37]. 

### 2.6. Genetic Associations That Require Validation

There are several other genes that may be associated with DLB. A potential protective variant within *PLCG2* (phospholipase C gamma 2) has been identified, yet this has not been replicated by subsequent GWASs [28,29,64]. Significant associations between DLB and variants within *CHRFAM7A* (CHRNA7-FAM7A fusion protein), *SCARB2* (scavenger receptor class B member 2), *BCHE* (butyrylcholinesterase), *PSEN1* (presenilin 1), and *NOS2* (nitric oxide synthase 2) have also been detected, although the evidence for these loci is conflicting and further replication is required [26,31]. The involvement of genetic variants within *LRRK2* (leucine rich repeat kinase 2) has been suspected, as the gene is implicated in Parkinson’s disease (PD) pathogenesis, but this evidence is inconclusive [65]. 

The first GWAS within DLB, published by Guerreiro and colleagues [28], discovered a significant association for the *BCL7C*/*STX1B* (BAF chromatin remodeling complex subunit BCLC7/syntaxin 1B) loci. This GWAS also identified the *GABRB3* (gamma-aminobutyric acid type A receptor subunit beta3) locus as being significantly associated with DLB, although no association was detected when the study was limited to include only pathologically diagnosed samples [28]. The GWAS also identified suggestive association of the *CNTN1* (contactin 1) locus that did not reach genome-wide significance [28]. Although, a second GWAS, published in 2019 by Rongve et al. [29], did not validate any of these findings and identified a novel suggestive association of the *ZFPM1* (zinc finger protein, FOG family member 1) locus. The lack of replication between the two GWASs may be due to differences in study design. The study by Rongve and colleagues included a substantially greater number of controls than the previous GWAS (82,035 vs. 4454), but included fewer cases (828 vs. 1743), of which none were pathologically diagnosed (0% vs. 76%). Validation with additional pathologically diagnosed cases is required to determine the true association of these loci.

## 3. The Transcriptomics of DLB

The study of DLB transcriptomics, which encompasses whole gene expression, transcript expression, and alternative splicing, is still a developing field. The published articles within the area, of which there are over 40 [25], have identified several pathways and genes of interest that may be pathogenic within DLB, as summarised in Figure 1.

### 3.1. Synuclein Aggregation

The aggregation of *α*-synuclein is a key component of DLB pathology. Prior genetic investigations have identified the involvement of *SNCA* and *SNCB* [27], and subsequent transcriptomic studies have further highlighted the potentially pathophysiological role of these genes within synuclein aggregation.

Increased *SNCA* expression has been suspected as a potential cause for *α*-synuclein aggregation for some time. A recent review, which analysed 31 studies that predominantly used quantitative polymerase chain reactions of candidate genes reported that the total expression of *SNCA* did not differ in post-mortem DLB brains when compared to controls [25]. Yet, biologically relevant changes of the transcriptome may be being driven at a transcript level. Alternative splicing, which is the variation of transcript ratios within a gene, may hide transcriptomic signatures from gene-level investigations.

*SNCA* has three main transcripts that arise from alternative splicing: *SNCA-98*, *SNCA-112*, and *SNCA-126* [43,66]. Multiple studies have identified upregulation of *SNCA-98* and *SNCA-112* in DLB brains when compared to controls [67,68,69]. Both *SNCA-98* and *SNCA-112* have a deletion of exon 5, which causes truncation of the C-terminus [70]. Shortening of the C-terminus produces variants with greater aggregation propensity [70]. The deletion also removes negatively charged amino acid residues, which increases the net charge and further promotes aggregation [70]. The upregulation of *SNCA-98* and *SNCA-112* may therefore promote synuclein aggregation. Conversely, *SNCA-126* downregulation has been detected in the prefrontal cortices and peripheral leukocytes of individuals with DLB [69,71]. *SNCA-126* has a deletion of exon 3, which shortens a region primarily involved in oligomerisation and aggregation [70]. As such, this transcript is associated with decreased synuclein aggregation [70]. The downregulation of this transcript, combined with the upregulation of transcripts that promote aggregation, highlights an alternative splicing mechanism that may trigger synuclein dysfunction, aggregation, and consequent Lewy pathology. 

Differential expression analysis of *SNCB* within DLB has also identified the potentially pathogenic signatures of individual transcripts. Two transcripts, *SNCB-tv1* and *SNCB-tv2*, displayed significant downregulation in the frontal and temporal cortices of DLB cases [72]. Considering *β*-synuclein is known to prevent α-synuclein aggregation [34], *SNCB* downregulation may be associated with the pathology of DLB through the dysfunction of *α*-synuclein anti-aggregation. Although, *SNCB-tv2* also displayed significantly increased expression in the caudate nucleus within DLB cases [72]. *SNCB-tv2* is distinct from *SNCB-tv1* as it lacks exon 2, and its upregulation may suggest that this exon causes gene dysfunction and DLB [72]. 

### 3.2. Protein Degredation

DLB is characterised by an accumulation of pathogenic proteins. The removal of such proteins is a typical process in healthy individuals, whilst DLB is associated with dysfunctional protein degradation. Two protein removal mechanisms, the autophagy lysosomal pathway (ALP) and the ubiquitin proteosome pathway (UPP), have been implicated in DLB following transcriptomic analysis [25].

The ALP degrades proteins and macromolecules utilising autophagosomes and lysosomes [73]. Target material is engulfed by an autophagosome and fused with a lysosome that inserts proteases and lipases to initiate degradation [73]. The ALP is thought to be the only mechanism capable of degrading aggregated proteins [73], and as such is important in DLB where numerous proteins aggregate into Lewy bodies. The downregulation of a gene involved within the ALP, *GBA*, has been observed within the substantia nigra of DLB brains [74]. Further research has also demonstrated that the associations of *GBA* may be driven by expression changes at a transcript level. The expression of *GBA-tv5* was found to be significantly downregulated in the temporal cortex of DLB brains, and downregulation of *GBA-tv1* was also found in the caudate nucleus and temporal cortex of DLB brains that presented with AD-related pathology [75]. *GBA* translates a key enzyme within the ALP, and downregulation of the gene and its transcripts likely results in ALP dysfunction [76]. This may prevent the degradation of aggregated proteins and precipitate DLB pathology [76]. 

The UPP is a second protein degradation system that has been implicated within DLB pathogenesis. The UPP is a mechanism responsible for the degradation of proteins within cells [77]. In healthy brains, the UPP tags damaged proteins with ubiquitin and facilitates their removal with proteosomes [77]. Downregulation of *UCHL-1* (ubiquitin C-terminal hydrolase L1), *PRKN* (parkin RBR E3 ubiquitin protein ligase), *SNCAIP* (synuclein alpha interacting protein), and *USP9Y* (ubiquitin specific peptidase 9 Y-linked), all of which translate to proteins within the UPP, have been identified in DLB [78,79,80,81]. The products of these genes contribute to protein tagging, protein degradation, and regulation of the UPP [78,79,80,81]. Reduced expression of these genes likely contributes to a dysfunctional UPP and protein degradation, and precipitates DLB pathogenesis.

### 3.3. Amyloid Deposition

The presence of A*β* fragments and aggregates is a common feature among neurodegenerative dementias, and individuals with DLB often exhibit amyloid co-pathology [50]. A*β*, which is a product of the amyloid precursor protein (APP), is deposited in over half of DLB cases [50]. 

The upregulation of *APP* transcripts *APP-770* and *APP-751* have been detected in the frontal cortices of DLB brains [82]. These transcripts both have a Kunitz protease inhibitory (KPI) motif and have also been shown to be upregulated in the cerebral cortex of DLB brains when compared to an *APP* transcript that lacks a KPI motif, *APP-695* [83]. These findings suggest the involvement of this motif within DLB pathology. The KPI motif is a 57 amino insert, which inhibits the activity of various proteases and prevent protein degradation [84]. KPI positive APP isoforms have been shown to increase amyloid deposition [84], and their elevated expression within DLB may explain the presence of amyloid pathology. 

Further expression analysis has also implicated the involvement of *BACE1* within amyloid deposition and DLB pathology. *BACE1*, which is translated to *β*-secretase [85], has been found to be significantly upregulated within DLB [86]. The *β*-secretase enzyme cleaves APP and initiates A*β* biogenesis [85], and its increased expression highlights a possible method of amyloid deposition. Additional studies have identified that *α*-synuclein promotes *β*-secretase processing of APP [87], suggesting that there may be a mechanistic link between *α*-synuclein aggregation and amyloid deposition.

### 3.4. Neuroinflammation

Chronic neuroinflammation has been identified as a prominent mechanism within several neurodegenerative disorders [88]. Its involvement within DLB is not well established, and the evidence from transcriptomic studies is conflicting. Transcriptomic analysis did not identify evidence of neuroinflammation within the pulvinar of DLB brains [89]. Further post-mortem investigations of the frontal cortex and the anterior cingulate cortex, and expression analysis of serum extracellular vesicles, have identified downregulation of neuroinflammation-associated genes, including several interleukins and chemokines such as *IL2* (interleukin 2), *IL6* (interleukin 6), and *CXCL2* (C-X-C motif chemokine ligand 2) [89,90,91]. It has been suggested that a downregulation of neuroinflammation-associated genes may cause neurodegeneration in DLB [25]. Decreased neuroinflammation may limit the brain’s ability to respond to DLB pathogenesis and increase the vulnerability of neurons [25].

However, other transcriptomic analyses have identified the increased expression of genes associated with neuroinflammation in DLB. Upregulation of pro-inflammatory cytokines, including *TNF* (tumor necrosis factor) and *IL6*, has been detected within the hippocampus and peripheral blood of individuals with DLB [92,93]. Downregulation of cell survival genes, such as *BDNF* (brain-derived neurotrophic factor), has also been detected within the hippocampus [93]. This may lead to neuronal vulnerability and an upregulation of MHC class II molecular expression that precipitates neuroinflammation [93]. These findings support the hypothesis that neuroinflammation, and the increased expression of genes associated with neuroinflammation, is associated with DLB. Neuroinflammation may induce apoptosis of neurons and interfere with cell signalling, triggering cognitive decline and protein aggregation within DLB [94].

A recent hypothesis is that neuroinflammation within DLB changes along the disease course [94]. Neuroinflammation may increase in mild and prodromal DLB and then may attenuate throughout disease progression [94]. This finding likely explains some of the variability in current transcriptomic analysis. Longitudinal studies are required to further investigate the development of cerebral inflammation and gene expression across the disease course.

### 3.5. Other Transcriptomic Signatures

Transcriptomic analysis has continued to make significant advancements in understanding DLB pathology. Over 1000 DEGs have been identified within people with DLB, and multiple studies have identified additional pathways and processes that are of interest.

Mitochondrial dysfunction is one such process that has been implicated within DLB pathology [25]. Upregulation of *CDKN2A* (cyclin-dependent kinase inhibitor 2A) has been detected in the prefrontal cortex of DLB brains, and this was correlated with decreased mitochondrial copy number [95]. *CDKN2A* is a cell-cycle inhibitor that induces cellular senescence [95]. Expression of this gene may limit mitochondrial replication and cause dysfunction through decreased energy production [95]. Further transcriptomic analyses have identified downregulation of mitochondrial genes *MT-ATP8* (mitochondrially encoded ATP synthase membrane subunit 8), *MT-CO2* (mitochondrially encoded cytochrome C oxidase II), *MT-CO3* (mitochondrially encoded cytochrome C oxidase III), and *MT-ND2* (mitochondrially encoded NADH:ubiquinone oxidoreductase) in the leukocytes of individuals with DLB [96] and decreased mitochondrial energy production in brains with Lewy body pathology [91]. 

The upregulation of the heat-shock proteins *HSP70* and *HSP27* has also been detected within DLB [97,98]. Heat-shock proteins may be involved in the removal of *α*-synuclein aggregates, and their upregulation may be a response to DLB pathology [99], or they may modulate immune response and be implicated in neuroinflammation [100]. Additional transcriptomic alternations have been discovered within solute carriers involved in synaptic neurotransmitter clearance, glutamate transport, and cell surface interactions [96,101]. 

Alternative splicing is also a process that may have a significant role within DLB pathology. Splicing of *SNCA* transcripts has been shown to facilitate synuclein aggregation [67,68,69,70], and a recent investigation also uncovered evidence of widespread dysfunctional alternative splicing within DLB [101]. Feleke and colleagues combined single-cell and bulk RNA sequencing to demonstrate that variations in transcript ratios are frequent across cell types within DLB [101]. Additional research is required to identify and investigate the genes that are alternatively spliced and play a role within DLB pathology.

Transcriptomic analysis has also identified possible biomarkers in the biological fluids of DLB cases. Analysis of blood mRNA expression within DLB has identified 17 DEGs that, if replicated, may hold diagnostic biomarker potential [81]. It has also been shown that the expression levels of *SNCA* transcripts in blood may be utilised to distinguish between DLB and other forms of dementia [69]. Further analysis has detected 37 qPCR verified DEGs in serum small-extracellular vesicles of DLB cases [89], demonstrating additional avenues for biomarker development. These signatures currently require additional replication and validation before being utilised as diagnostic biomarkers.

### 3.6. Transcriptomic Comparison with Other Dementias and Synucleinopathies

Despite overlapping clinical features, the molecular pathology DLB is distinct from other dementias and synucleinopathies. It is therefore important to identify molecular signatures that are unique to DLB. A recent post-mortem analysis detected widespread transcriptomic signatures between the brains of DLB, PD, and Parkinson’s disease dementia (PDD) cases [101]. The upregulation of *APOE* was identified within DLB brains when compared to PD and PDD, suggesting a greater involvement of amyloid pathology [101]. *UCHL-1*, which translates to a key protein within the UPP, was downregulated when compared to PD and PDD [101]. This finding highlights that UPP dysfunction may play a more prominent role within DLB pathology. A comparison between DLB and AD blood mRNA identified 18 DEGs, and subsequent pathway analysis suggested that interferon response was upregulated in AD [81]. The expression of cholinergic receptors, *CHRM1* and *CHRM4*, have also been shown to be upregulated in AD when compared to DLB [102]. Further research has demonstrated that the transcriptomic differences between AD and those with Lewy pathology is dependent on brain region [103]. Transcriptional dysregulation appears to correlate with neurodegeneration, and individuals with Lewy pathology are likely to experience dysregulation in the substantia nigra, whilst cases of AD are more likely to show dysregulation in the parietal lobe [103]. Further research is warranted to determine how DLB differs from other dementias and synucleinopathies across the brain region to improve molecular understanding and facilitate accurate diagnosis.

## 4. Opportunities for Data Science in DLB

Existing genomic and transcriptomic analyses have greatly improved our understanding of DLB. It has revealed disease pathways such as synuclein aggregation and protein removal [25], and identified the involvement of genes such as *SNCA*, *APP*, and *GBA* [26]. Yet, numerous research gaps remain. Data science, and the computational analysis and of large datasets, has the capability to provide insight into these areas. A summary of these research areas and their available tools is presented in Table 2. Genomic data could be used to facilitate the early administration of disease-modifying therapeutics. There is also a demand for more comprehensive transcriptomic analysis that investigates transcript-level signatures across the entire transcriptome. This, in addition to multi-omic analysis that combines genomic and transcriptomic data to identify factors hidden from single-omic analysis, will increase the understanding of DLB pathology and facilitate improved drug development.

### 4.1. Genetic Prediction for Early Case Identification

DLB is a degenerative condition, and disease pathology is currently irreversible. As such, the disease-modifying therapeutics that are currently in development are focussed on slowing disease progression [127,128]. There is increasing emphasis on the early identification of cases so that disease-modifying interventions can be used in a timely manner when they are developed before the significant progression of pathology. Current clinical methods that use the presence or absence of symptoms to diagnose cases are not compatible with early intervention. Pathology can begin more than 15 years prior to the onset of dementia [129], and the average survival time after diagnosis is just over 4 years [20]. Symptomatic identification does not recognise cases early enough to facilitate relevant disease-modifying therapeutics. Genetic prediction methods, developed by data science, could be used to identify individuals at risk of DLB before symptoms develop. Genetic material is stable, easily obtainable, and cheap to analyse, which suggests that genetic prediction may become a feasible method of early case identification.

The most common computational technique used in genome-wide prediction is polygenic risk score (PRS) analysis [130,131]. PRSs use risk predictions from a training dataset to estimate individual risk in a test dataset [130]. Within PRS analyses, these two datasets are termed the base and the target. A base dataset containing the effect scores for SNPs is constructed from GWAS summary statistics [130]. The target dataset contains the genotype and disease phenotype information for individuals of interest [130]. The PRS protocol attributes effect scores from the base dataset to the genotypes within the target dataset and combines them to produce a cumulative risk score for each individual [130]. The cumulative scores and phenotypes for the individuals are then compared to determine how accurate the model is in discriminating between case and control [130]. Typical PRS procedures apply p-value thresholds to limit the inclusion of variants that will have reduced predictive accuracy [130].

Genomic prediction has been used extensively within AD with an accuracy of over 80% [131,132,133]. Yet, the use of genetic prediction within DLB is limited, as sufficiently powered GWASs have only recently been conducted. One investigation utilised only GWAS-significant markers to implement PRS analysis within DLB [134]. It demonstrated that scoring from five variants had a relative risk ratio of 3.22 (95% confidence interval: 1.62–6.40) for DLB and attributed significantly greater risk values to cases when compared to controls (*p* < 0.001) [134]. Although, previous analysis within AD has revealed that PRS accuracy can be increased from 70.0% to 74.1% by including variants beyond the GWAS-significance threshold [135]. There is a clear requirement for genetic risk analysis of DLB that utilises variants beyond a GWAS-significance threshold to determine the maximum accuracy with which cases can be stratified.

Recent evidence indicates that supervised machine learning models can greatly improve the accuracy of genomic prediction when compared to PRSs [136]. A supervised deep neural network increased predictive accuracy from 61.6% to 67.3% when compared to typical PRS computational analysis within the genetic data of breast cancer patients [136]. Supervised machine learning models are data-intensive approaches that utilise annotated training datasets to uncover hidden structures within the data and predict risk within a test dataset [137]. Conventional approaches, such as support vector machines and convolution neural networks, have been used previously for classification problems in other domains [137]. Although, the large number of features in genetic data creates general overfitting problems in proposed models, limiting the accuracy of prediction [137].

Future investigations should aim to apply both supervised machine learning and PRS analysis to assess the accuracy of genetic prediction within DLB. If accurate genetic prediction can be achieved, this analysis may hold significant value for the future management of DLB. Disease-modifying treatments, coupled with accurate and early case identification, may facilitate effective intervention [127,128]. Neflamapimod may progress to phase III trials in the near future, but the heterogenous nature of DLB still poses a sizeable challenge [128]. 

### 4.2. Transcsriptome-Wide Gene Expression Analysis

The molecular pathology of DLB remains unclear, and the development of disease-modifying therapeutics has been modest and slow [21]. Focus on previously identified pathways has yielded inconsistent results, as there is a sizeable proportion of the pathology yet to discover [21,128]. Existing transcriptomic analysis of DLB has revealed significant transcript-level alterations, although most of these investigations focussed on candidate genes and were limited in scope [25,101]. Transcriptome-wide identification of alternatively spliced genes and differentially expressed transcripts (DETs) has not been conducted. This analysis may reveal significant transcript-level alterations in novel pathways, which may increase our understanding of DLB pathology and facilitate improved development of disease-modifying therapeutics. Data science provides an opportunity to conduct transcript-level expression analysis across the entire transcriptome, simultaneously.

Transcriptome-level analysis begins with the quantification of RNA abundance. Following RNA-sequencing and quality control, the base sequences of reads are typically compared to that of a reference genome to align each read to its genomic location [138]. The abundance of mapped reads at each transcript is then calculated. Recently developed packages have implemented quasi-alignment, which is a fast and efficient alternative to previous techniques and removes the need for genome alignment [139]. Quasi-alignment compares reads to a genome index, and the abundance of each transcript is estimated based upon the number of overlapping reads [139]. Subsequent DET analysis then compares the normalised counts of each transcript between study groups to determine the statistical significance of differential expression [114]. DET analysis typically employs exact tests with no degrees of freedom and adjusts for transcript length [114]. Alternative splicing analysis also utilises RNA-abundance calculations but analyses the different ratios of transcript expression within genes and across conditions [140]. The abundance of each transcript is calculated and presented as a proportion of quantified transcripts within its gene. Typical alternative splicing analysis first investigates whether there is significant deviation in proportions between conditions and across all transcripts within the gene [118]. Subsequent calculations then determine which transcripts are contributing to this finding [118].

Transcript-level DET and alternative splicing analysis has revealed novel molecular signatures within AD [141,142]. DET analysis revealed over 2000 upregulated and downregulated transcripts in asymptomatic and symptomatic AD samples [142]. Alternative splicing analysis conducted on the same samples identified a further 1200 differentially spliced events in individuals with AD [142]. Additional splicing analysis in AD revealed novel differential transcript usage in ADAM10, BIN1, CLU, and TREM2 [141]. It is apparent that DET and alternative splicing analysis methods have the potential to identify novel associations, particularly within DLB, which is known to have wide-spread signatures at a transcript level [101].

Further developments within data science may also accelerate the identification of transcript-level associations in the near future. Transcriptome-wide association studies (TWASs) can be used in the absence of transcriptomic data or if existing analysis is underpowered. This is particularly relevant within DLB research, as existing RNA-sequencing studies conducted on post-mortem material are yet to include more than 10 DLB samples [25,101]. A TWAS leverages the significant statistical power of GWASs and public expression quantitative trait loci (eQTL) databases to predict transcriptomes and associate them with disease outcomes [143]. It can either use individual-level data, where the genotype information of each sample is compared to the eQTL database to estimate and associate expression at each loci, or it can use summary-level data, where standardised effect scores from GWASs are multiplied by the predicted gene expression effect for each variant [143]. Through both forms, TWASs have the potential to identify novel signatures within disease pathology. It has previously been used to identify the potential involvement of 50 novel genes within AD [144], and there is hope that its use within DLB may increase the understanding of disease pathology, facilitating accelerated drug development.

The application of these data science methods, and the transcriptome-wide identification of DETs and alternatively spliced genes, will advance the understanding of DLB pathology. This may identify novel pathogenic pathways and processes that can be targeted for therapeutic intervention. The identification of these targets has the potential to accelerate drug development and advance the use of disease-modifying therapeutics. This area represents a discipline with significant promise, and it should be the subject of increased scientific attention.

### 4.3. Unsupervised Machine Learning for Multiomics

The unclear understanding of DLB pathology may also be caused by the lack of multi-omic analysis. DLB is often caused by a combination of domains within the biological dogma [145], and analysis that only considers one domain is likely to overlook pathological components that span multiple omic datasets. For example, genomic alterations may be significant only in the presence of transcriptomic factors. To date, the genomics and transcriptomics of DLB samples have been investigated independently, and no studies have combined the two with the analysis of matched samples. Data science facilitates unsupervised machine learning for multi-omic analysis, which may provide insight into DLB and identify novel pathways and molecular signatures. This is likely to improve the molecular understanding of DLB, potentially reveal therapeutic targets, and facilitate improved drug development. 

Unsupervised machine learning, unlike supervised machine learning, utilises unlabelled data to reveal hidden associations and structures within datasets [145]. Unsupervised multi-omic analysis can be categorised into regression-based, clustering-based, or network-based methods [145]. Regression-based methods identify associations between layers of omics data to identify latent factors that may be implicated in disease pathology [145]. Regression-based methods typically reduce the complexity of datasets whilst preserving key biological drivers within the data [123]. These determine the extent to which latent factors account for variation within the omics datasets [123]. Clustering-based methods identify groups or modules within omics datasets to identify disease sub-types [145]. They generally create a sample-by-sample similarity matrix for each omics dataset before fusing networks to combine the matrices and produce a single similarity matrix [125]. Network-based multi-omic methods leverage information from existing datasets to create networks that depict the functional relationships within the datasets [145]. Network-based approaches can first utilise weighted gene co-expression analysis to establish gene modules and nodes [126]. Features from additional omics datasets are then added to provide combined scores for the modules, and gene ontology can be utilised to facilitate enrichment analysis [126]. 

Unsupervised multi-omic analysis has been successful at identifying novel associations in AD and oncology. Genomics, transcriptomics, and other omic datasets have been combined to identify subtypes of amyloid pathology and key biological nodes within AD cases [146]. Further analyses have identified novel associations within pathways of amyloid pathology, neuronal injury, and tau hyperphosphorylation from the CSF of AD cases [147]. Similar analyses within oncology have identified molecular features associated with tumour invasiveness [148], immunotherapy susceptibility [149], and the subtypes of oncogene mutations [150]. Emerging methods are also supplementing unsupervised models with labelled datasets through semi-supervised learning to improve the identification of novel signatures. Semi-supervised learning has been shown to be cost effective and accessible and has been successful in mammogram classification [151].

Unsupervised multi-omic analysis has provided insight into the molecular pathology of other diseases, and it has the potential to do the same within DLB. Future studies should implement unsupervised machine learning to improve disease understanding and identify signatures that singular omic analyses have failed to recognise. These findings may accelerate the development of disease-modifying drugs and facilitate therapeutics that limit the progression of DLB.

## 5. Conclusions

DLB is a significant public health issue. The molecular pathology of DLB is unclear, and additional research is required to facilitate the development of therapeutics and diagnostic biomarkers. Genomic analysis has identified numerous molecular signatures. Variants within *SNCA*, *APOE*, *GBA*, *SNCB*, *SNCG*, and *MAPT* have all been associated with DLB [26]. Two GWASs have identified numerous variants that require additional validation [28,29]. 

Transcriptomic analysis has also identified the involvement of several pathways and processes. Differentially expressed genes and transcripts have been detected within pathways such as synuclein aggregation, protein degradation, amyloid deposition, and neuroinflammation [25]. Transcriptomic studies have also identified the involvement of mitochondrial dysfunction and wide-spread alternative splicing [25,101].

Data science and its future use has the potential to provide insight into the research gaps within DLB. Data science and deep learning methods will facilitate accurate genomic prediction to stratify individuals with a high risk of DLB. Methods that investigate transcript-level alterations across the transcriptome can also be utilised to identify important DETs or alternatively spliced genes that may play a significant role within DLB pathology. The application of TWASs may detect novel signatures following an increase in statistical power. The use of multi-omic analysis will detect novel molecular signatures and advance the understanding of DLB. 

Future research should not only focus on the application of these techniques but also advancement. Machine learning analysis likely holds the most potential. Development of supervised, semi-supervised, and unsupervised machine learning methods will facilitate improved genomic prediction for case stratification and the identification of novel molecular signatures that may accelerate drug discovery. This will aid the early detection and treatment of DLB and may reduce the substantial and ever-increasing disease burden.

## Figures and Tables

**Figure 1 cells-13-00223-f001:**
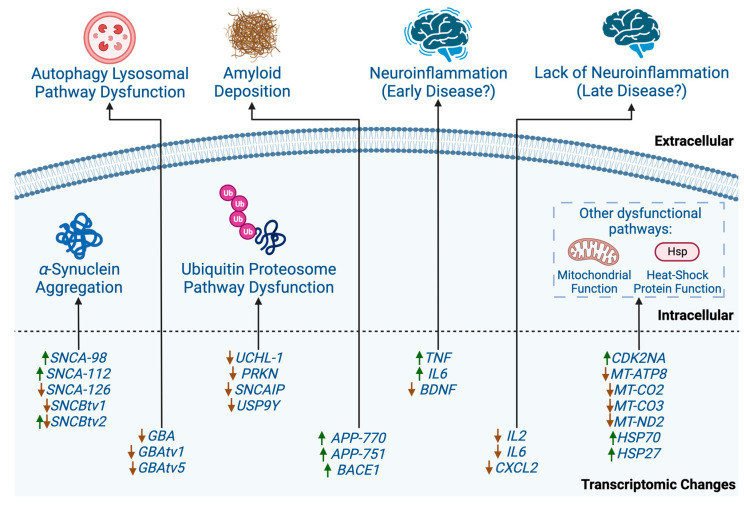
An overview of the dysfunctional pathways within DLB, both intracellular and extracellular, that have been identified by transcriptomic analysis. The gene expression changes of relevant genes and transcripts have been included to show how upregulation and downregulation may play into the dysfunction of each pathway. Green arrows within the transcriptomic changes indicate an increase in expression. Red arrows within the transcriptomic changes indicate a decrease in expression. Green arrows combined with red arrows indicate that expression can be upregulated or downregulated, depending on the brain region. Created with https://app.biorender.com (accessed on 14 December 2023).

**Table 1 cells-13-00223-t001:** Genes, with selected variants, that have displayed association with DLB in two or more studies.

Gene	Variant	Effect Size (95% Confidence Interval)	Sample Size (Cases:Controls)	Study
*SNCA*	rs7681440	0.73 (0.66–0.81)	1743:4454	Guerreiro et al. [28]
	rs1348224	0.71 (0.61–0.83)	922:971	Guella et al. [30]
	rs894280	0.75 (0.67–0.85)	788:2624	Bras et al. [31]
	rs2301135	1.40 ^a^	720:6490	Rongve et al. [29]
*GBA*	rs35749011	2.55 (1.88–3.46)	1743:4454	Guerreiro et al. [28]
	“Mutation carrier status”	8.28 (4.78–14.88)	721:1962	Nalls et al. [32]
	“Pathogenic GBA mutations”	7.60 (1.80–31.90)	79:391	Tsuang et al. [33]
	rs2230288	5.57 ^a^	720:6490	Rongve et al. [29]
*APOE*	rs429358	2.40 (2.14–2.70)	1743:4454	Guerreiro et al. [28]
	rs429358	2.28 ^a^	720:6490	Rongve et al. [29]
	rs769449	2.79 (2.40–3.24)	788:2624	Bras et al. [31]
	ε4 haplotype	2.50 (2.29–2.70)	922:971	Guella et al. [30]
*SNCB*	rs11739753	0.63 (0.44–0.90)	172:97	Nishioka et al. [34]
	V70M	N/A ^b^	33:660	Ohtake et al. [35]
*MAPT*	H1G haplotype	3.30 (1.34–8.12)	442:2456	Labbé et al. [36]
	G86S	N/A ^b^	1118:432	Orme et al. [37]
	H1 haplotype	1.81 (1.05–3.14)	51:325	Cervera-Carles et al. [38]

^a^ Confidence interval not published. ^b^ Investigation identified presence of variant in one or more cases of DLB.

**Table 2 cells-13-00223-t002:** Areas of opportunity for data science in DLB, and an overview of commonly used tools.

Research Area	Data Science Tool	Description
Genomic Prediction	PLINK [104]	Genome analysis toolkit that includes scoring functions.
	PRSice-2 [105]	Automated scoring package that performs sequential threshold testing.
	LDpred-2 [106]	Incorporates linkage disequilibrium within genetic scoring to improve accuracy.
	Lassosum [107]	Utilises penalised regression and linkage disequilibrium within genetic scoring to improve accuracy.
	Tensorflow [108]	Machine learning system that facilitates application of supervised models.
RNA Alignment and Quantification	HISAT2 [109]	Alignment tool that utilises hierarchical indexing.
	STAR [110]	Alignment tool that utilises sequential search models.
	FeatureCounts [111]	Read summarisation package that counts reads within aligned data.
	Salmon [112]	Pseudoalignment package that quantifies reads without alignment.
	Kallisto [113]	Pseudoalignment package that quantifies reads without alignment.
Transcriptome-level Expression Analysis	edgeR [114]	Employs Poisson and empirical Bayes models to calculate differential expression.
	DESeq2 [115]	Utilises shrinkage estimation for differential expression analysis.
	Cufflinks [116]	RNA analysis package that includes Cuffdiff, which assesses splicing from aligned reads.
	Leafcutter [117]	Determines differential intron usage within annotation-free read data.
	DRIMSeq [118]	Analyses differential transcript usage within annotation-free read data.
Transcriptome-wide Association Analysis	PrediXcan [119]	Estimates and associates gene expression from cis-acting variants within a single tissue.
	MultiXCan [120]	Estimates and associates gene expression from cis-acting variants within multiple tissues.
	BGW-TWAS [121]	Estimates and associates gene expression from cis and trans-acting variants.
	MOSTWAS [122]	Incorporates multi-omic data and distal variants to estimate and associate gene expression.
Unsupervised Machine Learning for Multi-omic Analysis	MOFA [123]	Regression-based method to integrate multiple omic datasets and identify latent factors.
	DIABLO [124]	Regression-based method to integrate multiple omic datasets and identify latent factors.
	Similarity Fusion Network [125]	Clustering-based model that combines multiple omic datasets to identify relationships between samples.
	Lemon-Tree [126]	Network-based technique that incorporates multiple omic datasets and ensemble methods for network inference.

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
