# Peer review of "Dementia with Lewy Bodies: Genomics, Transcriptomics, and Its Future with Data Science"

_cells, 2024, doi:10.3390/cells13030223_

Round 1

Reviewer 1 Report

Comments and Suggestions for Authors

A very interesting and useful Review, from the genomics and transcriptomics point of view, offering some hope in the possible early diagnosis and possible interventions in DLB

What I missed, and I think it will give more weight to the review paper, is some current neuropathology on DLB about genomics and transcriptomics. When, and where (59) are these hallmarks, the Lewy bodies, detected and what is the result on the brain of the affected individuals? It may shed some light on the early diagnosis and subsequent treatment.

DLB is perhaps dementia with more severe Neuropsychiatric symptoms, and would be important in this Review to give some information in that respect, particularly besides (13) "the socioeconomic impacts". Pe the tremendous impact on caregivers and others, the emotional burden etc.

Reviewer 2 Report

Comments and Suggestions for Authors

The narrative review Dementia with Lewy Bodies: Genomics, Transcriptomics, and Its Future with Data Science, by Goddard, Brookes, et al., is clear, well written and comprehensive. Particularly, it demonstrates the relationship between pathology, genomics and transcriptomics, showing the possibilities offered by the new data science approach. As a whole, I find the manuscript original and interesting; however, I have some suggestions for improving it:

1) Introduction - Page 2, lines 49-50: The authors should point out that this is an underestimate, since considering studies with a defined neuropathological diagnosis the percentage rises to 20%, and up to 30% considering mix AD/LBD pathology. Please, see and quote: Oinas et al. Reappraisal of a consecutive autopsy series of patients with primary degenerative dementia: Lewy-related pathology. APMIS 2007; 115: 820-7. doi: 10.1111/j.1600-0463.2007.apm_521.x. APMIS. 2007. PMID: 17614849.

2) Introduction - Page 2, lines 57-58:  It is too simplistic the concept that synuclein aggregation is the cause of the disease. Indeed, protein aggregation is a clue, a signature that represents the consequence and not the cause of the disease, which is probably due to alterations that occur upstream. Precisely for this reason, omic sciences are important; they may reveal the pathologic mechanisms that precede protein aggregation. I suggest modifying this sentence accordingly.

3) Introduction - Page 2, lines 58-59: Please, don’t forget Lewy neurites; I’d suggest to change the sentence: “Lewy bodies and Lewy neurites, the pathological hallmarks of DLB and other Lewy type synucleinopathies (PD, PDD, LBD)". Please, see and quote: Beach, T.G., et al. Unified Staging System for Lewy Body Disorders: Correlation with Nigrostriatal Degeneration, Cognitive Impairment and Motor Dysfunction. Acta Neuropathol. 2009, 117, 613–634.

4) Section 2.1. SNCA - Page 3, line 92: Since the significance of synuclein aggregation is not known, it would be better to say, "they are associated with...", rather than "occur following..."

5) Section 2.2. GBA - Page 3, line 123: to clarify the concept may be useful to add: "due to a reduced clearance from brain parenchyma".

6) Section 3.1. Synuclein Aggregation - Page 7, line 250: I'd prefer to read something like: "trigger synuclein dysfunction, and in turn Lewy type pathology including Lewy bodies and Lewy neurites". Protein aggregation is the consequence, the final outcome of a cascade of events, and not the primary cause of the disease, which starts upstream. In turn, proteinopathies may propagate and exert a neurotoxic effect, but probably the beginning of neurodegeneration is upstream. The authors should discuss these issues.

7) Section 3.5. Other Transcriptomic Signatures - page 9, line 351: CDKN2A gene corresponds to cyclin-dependent kinase inhibitor 2A. All the acronyms should be spelled out; I suggest preparing a table for this purpose.

8) Section 3.6. Transcriptomic Comparison with Other Dementias and Synucleiopathies - Page 9: This section is a bit confusing and poor, and should be enriched. I suggest discussing that AD and LBD appear as two clearly distinct diseases from a molecular, transcriptomic and topographical point of view, despite the presence of a partial overlap in neuropathology and gene deregulation. Please, see and quote the interesting work of Palmieri I et al., showing a comprehensive clinical, neuropathological and transcriptomic comparison between AD and LBD, and a control case. (Palmieri I, et al. Differential Neuropathology, Genetics, and Transcriptomics in Two Kindred Cases with Alzheimer’s Disease and Lewy
Body Dementia. Biomedicines 2022; 10: 1687https://doi.org/10.3390/biomedicines10071687).

9) Section 4.1. Genetic Prediction for Early Case Identification - page 12, lines 465: I think there is an excessive optimistic emphasis in this sentence. Moreover, MacDonald (ref.121) states that "The current treatment options focus on relieving symptoms; no disease-modifying therapies are available". Furthermore, It should be considered that, at present, only phase 2 trials are underway (Neflamapimod will enter phase 3). Of course, from a research point of view it is all very interesting, but it must be taken into account that it may be very harmful to know well in advance that you have a high probability of developing a disease for which there is no cure. These aspects should be discussed more objectively.
